# Bioavailability Assessment of Metals from the Coastal Sediments of Tropical Estuaries Based on Acid-Volatile Sulfide and Simultaneously Extracted Metals

Ana Paula de Castro Rodrigues [1,2], Matheus Marinho Pereira [1], Aline Campos [3], Tássia Lins da Silva Quaresma [1], Rodrigo Pova [1], Thatianne Castro Vieira [1,4], Rút Amélia Diaz [1], Manuel Moreira [1], Denise Araripe [5], Christiane do Nascimento Monte [1,6,*] and Wilson Machado [1]

[1] Program of Geochemistry, Department of Geochemistry, Fluminense Federal University (UFF), Outeiro de São João Batista s/n, Niterói 24020-141, RJ, Brazil

[2] Health Science Center, Rio de Janeiro Federal University (UFRJ), Av. Carlos Chagas Filho, 373, Cidade Universitária, Rio de Janeiro 21941-902, RJ, Brazil

[3] Integrated Faculties Maria Thereza Oceanography and Marine Biology, Av. Visconde do Rio Branco, 869, São Domingos, Niterói 24210-006, RJ, Brazil

[4] Pos-Graduation Program on Oceans and Earth Dynamics, Federal Fluminense University (UFF), Av. Milton Tavares de Souza s/n, Gragoatá, Niterói 24210-340, RJ, Brazil

[5] Program of Chemistry, Departament of Chemistry, Fluminense Federal University (UFF), Out. de São João Batista s/n, Niterói 24020-141, RJ, Brazil

[6] Geology Department, Federal of Western of Pará (Ufopa), Av. Vera Paz s/n, Salé, Santarém 68040-255, PA, Brazil

* Correspondence: christiane.monte@yahoo.com.br

**Abstract:** Bioavailability assessment is important for evaluating the risks to the local biota, and the combined use of several ecological risk indices in eutrophic environments allows the best analysis of the local reality for decision-making. The relationship between acid volatile sulfide (AVS) concentrations and simultaneously extracted metals (SEM) allows us to infer the metal holding capacity of sediment, with the objective of evaluating the potential bioavailability of trace metals (Cd, Cu, Ni, Pb, and Zn) using ecological risk indices, such as the ΣSEM/AVS model and Adverse Effect Index (AEI), in surface sediments from Guanabara Bay and Sepetiba Bay, Brazil. AVS was determined using a colorimetric method and SEM with ICP-OES. In general, almost all sampling in Sepetiba Bay showed ΣSEM/AVS ratio values above 1. However, all results for the ΣSEM/AVS ratio found for the Guanabara Bay sediments were <1 for both estuaries. After normalization by organic carbon content, a possible toxicity risk for biota was found in Sepetiba Bay. However, the AEI in Guanabara Bay was above 1 for all metals in most samples, also indicating a risk to the biota.

**Keywords:** AVS; ecological risk; adverse effect index; Guanabara Bay; Sepetiba Bay

## 1. Introduction

Sediments are formed by the deposition of suspended particles from rivers, lakes, estuaries, and oceans. They function as important compartments of aquatic ecosystems and serve as habitats for many species of organisms [1,2]. They consist of minerals of varying sizes including clay, silt, and sand. Sediments in marine environments are the main reservoirs of organic and metallic pollutants [3,4].

Among the mechanisms influencing the retention of metals, the generation of insoluble metal sulfides has been recognized as a key process in determining the behavior of these elements in estuarine sediments. Due to their high adsorptive capacity, sediments have characteristics that favor the accumulation of trace metals. High amounts of organic matter and sulfides can fix metal ions as insoluble sulfides, and organic substances form organometallic compounds [5]. In areas impacted by anthropic action, sediments can not

only be a pollutant storage from the water column, but also a source of pollutants because of the possibility that these contaminants are released back into the environment [6,7].

Under reducing conditions, metals can be incorporated into the solid phase of the sediment to form amorphous monosulfides, operationally defined as volatile sulfides, in reaction with an acid (acid volatile sulfides, AVS). The formation of AVS is related to the anoxic conditions of the sediment and the content of labile organic matter and sulfate, which leads to a higher activity of sulfate-reducing bacteria [4,8]. According to Di Toro et al. (1990), insoluble metal sulfide complexes regulate the metal concentration in the pore water of the sediment, due to the formation of AVS.

These sulfides may also play important roles in controlling the bioavailability of metals in anoxic sediments. Research has established that the relative amount of simultaneously extracted metals (SEM) and AVS is important in predicting bioavailability [4,9,10]. If the molar amount of SEM is less than the amount of AVS, the toxic metals in the sample will potentially not be bioavailable in interstitial waters, as there is enough AVS to react with these metals and remove them from the solid-phase solution [11].

This ratio has been recognized as a key process for determining trace metal behavior in coastal sedimentary environments where anaerobic conditions occur [2,8,12]. This function of sulfides is related to the incorporation of available metals into the dissolved phase by the solid phase of sediments through the formation of metal monosulfides (or the adsorption of metals to them), predominantly composed of FeS [11,13].

The probability of transfer of potential toxic elements (PTE) to aquatic and benthic organisms is usually verified by the ratio of simultaneously extracted metals to acid volatile sulfides ($\Sigma$SEM/AVS) [14,15]. If this ratio is smaller than or equal to one, the sample contains more sulfide, which can precipitate PTE.

Sepetiba Bay has a widely recognized history of anthropogenic metal contamination [16,17]. It is also noteworthy that the anthropogenic impacts on the drainage basins and within the bay itself are expanding, including the effects of industrial activities, intensification of urban occupation, and implementation of more port activities, generating increasing environmental concern over the coming decades [18].

Guanabara Bay has been identified as one of the most polluted coastal environments in Brazil [19,20]. Since the 1950s, the degradation process has intensified, with the emergence of one of the largest industrial complexes in the watershed region. The industrial complex comprises diverse industries, such as gas and oil, pharmaceuticals, and metallurgy. Furthermore, the bay surrounds the largest oil refinery in the country and the former Gramacho landfill, which was considered the largest in South America until 2012, when it was closed [2,20]. In addition, high urban growth in the state's metropolitan region has resulted in intense coastal contamination, especially by metals, which are one of the main by-products of industrialization. In recent decades, several initiatives have been undertaken to characterize and provide a behavioral understanding of these contaminants in Guanabara Bay [2,20–22].

Understanding the dynamic bioavailability of metals in estuaries in densely urbanized areas is a crucial issue concerning metal contamination, particularly in areas that are used for fishing, such as Guanabara Bay and Sepetiba Bay. However, most studies have mainly focused on one area. The discussion on bioavailability and toxicity of metals in sediments using SEM-AVS models in two types of estuaries, one eutrophic and the other mesotrophic, is poorly discussed in an integrated way, which leaves a gap in the literature. Thus, this study contributes to the understanding of the geochemical characteristics and dynamics of AVS in sediment and its relationship with some factors that regulate bioavailability, such as organic matter [2,8].

Assessment of potential bioavailability in eutrophic environments is complex due to the high level of degradation. In such cases, the combined use of several assessment tools is necessary to obtain a robust result according to the local reality before decision making by environmental agencies. Therefore, the present study aimed to evaluate the potential bioavailability of trace metals (Cd, Cu, Ni, Pb, and Zn) using the relationship

between metals, AVS, and organic carbon present in the surface sediments of Sepetiba and Guanabara Bays, RJ, Brazil, both historically known for environmental impacts related to trace metal contamination. Two tools were used to achieve this objective the ΣSEM/AVS model and a biological adverse effects index at the two estuaries.

## 2. Material and Methods

### 2.1. Study Area

Sepetiba Bay (SB) is a coastal system with port activities of great socioeconomic importance for the state of Rio de Janeiro, with a population of over 1.8 million, in its drainage basin, which also has agricultural activities [18,23]. The bay stands out as one of the most metal-impacted environmental areas (particularly Cd and Zn) in the country due to a large industrial point source–a pile of ore residuals (rich in Pb, Cd, Zn, and Cu) disposed in the north of the bay, near Madeira Island (MID) [18,24]. Although this pile was recently removed and a decontamination project was implemented, the ore residuals of an old zinc coprocessor (currently deactivated) were lixiviated for decades, reaching the Sepetiba Bay waters, especially after intense rainfall events [18,23].

The San Francisco Channel (SFC) is the main source of freshwater to Sepetiba Bay, but it suffers from metal contamination, such as Zn, Cd, Pb, and Hg, originating from the Santa Cruz industrial park, which industries a metallurgical plant, along with the Paraíba do Sul River [23,25].

Guanabara Bay (GB) is one of the largest urban bays in the world and suffers from discharge of untreated industrial and domestic sewage [7]. This ecosystem is considered eutrophic to hypereutrophic, due to the high organic load [7,20,26]. In addition, contamination by metals such as Cr, Cu, Hg, Pb, and Zn in the northwestern region of the bay has been reported [2,20,27].

The Iguaçu and Meriti rivers are the main rivers that flow into the northwestern region of the GB, contributing freshwater input to this region [20]. These rivers drain industrial complexes located in the metropolitan region of Rio de Janeiro.

The largest oil refinery in Brazil, the Duque de Caxias Refinery (REDUC), is located at the mouth of the Iguaçu River (IR). The Iguaçu-Sarapuí River system receives a significant amount of untreated domestic wastewater due to the absence of sanitation [20,28]. The estuarine region of the Iguaçu River is silted, frequently showing areas with a maximum depth of 50 cm during high tide.

The Meriti River basin (MR), Guanabara Bay, RJ, drains the Baixada Fluminense region and receives high loads of domestic and industrial waste containing heavy metals, of which Hg is highly toxic [2,29]. In addition, the Meriti River mouth routinely faces problems associated with siltation, hindering the circulation and renewal of the bay waters, as well as navigability [2].

### 2.2. Sampling

Four sampling campaigns were conducted (Figure 1): two in Sepetiba Bay (one in the Madeira Island region on 6 July 2012(Figure 1E), and one in the São Francisco Channel on 11 August 2012 (Figure 1F) and two in Guanabara Bay on the Meriti River estuary on 21 March 2012 (Figure 1D), and Iguaçu River estuary on 30 April 2011 (Figure 1C) at 48 collection stations (12 in each area). Surface sediments were obtained with the aid of a deep van Veen search, and the samples were stored in glass jars with lids and frozen immediately after collection. Physicochemical parameters (pH, salinity, conductivity, and dissolved oxygen) of the surface water were measured in situ using a YSI model 85 multitrack.

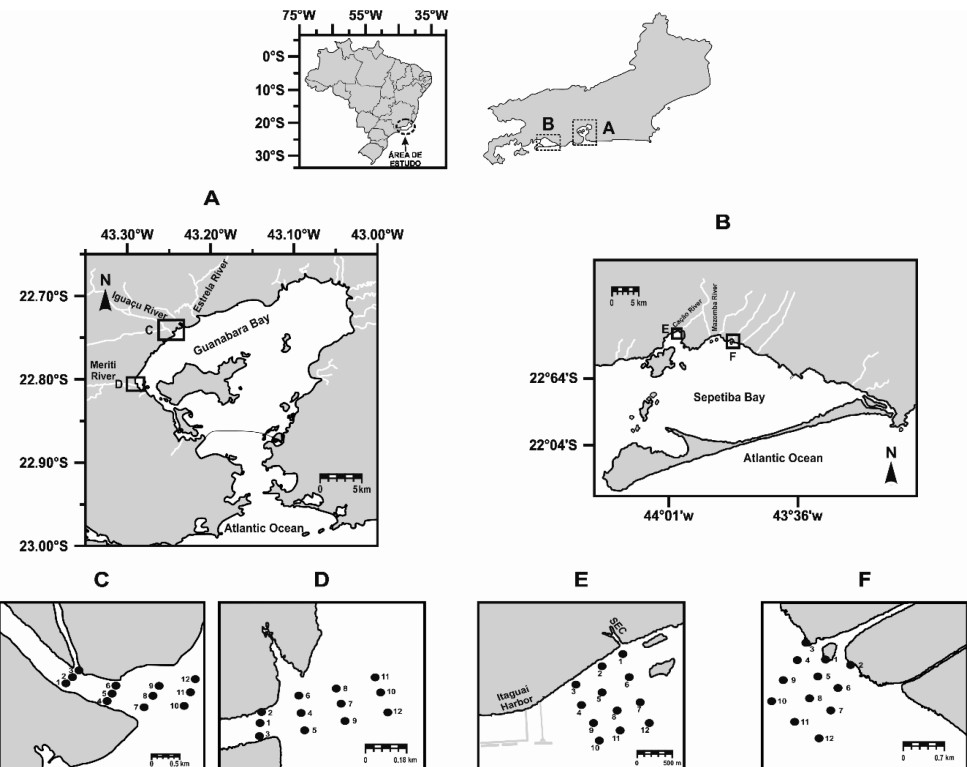

**Figure 1.** Sampling locations. (**A**): Guanabara Bay; (**B**): Sepetiba Bay; (**C**): Iguaçu River; (**D**): Meriti River; (**E**): Madeira Island (**F**): São Francisco Channel.

### 2.3. Acid Volatile Sulfides (AVS)

The extraction and determination of AVS concentrations were performed according to [9]. Wet subsamples of the sediments were subjected to acid distillation with 6 M HCl, and the sediment samples were purged in a closed system with argon for 10 min to release free hydrogen sulfide ($H_2S$). Subsequently, 20 mL of 6 M HCl solution was added, mixed with the sediment, and stirred for 1 h. The $H_2S$ released was collected in a solution of 0.5 M NaOH. The sulfides were determined in an absorbent solution by spectrophotometry (UV-VIS Spectrophotometer, SHIMADZU model UV-1601) using the colorimetry method [30].

### 2.4. Determination of Simultaneously Extracted Metals

Acid suspensions were filtered and used to determine the concentrations of simultaneously extracted metals (SEM) [9]. The concentrations of Fe, Mn, Cd, Cu, Ni, Pb, and Zn were determined by sequential radial vision plasma source optical emission (ICP OES) spectrometry, Horiba Jobin Yvon, Ultima 2 model (Longjumeau, France), equipped with cyclonic nebulizer chamber, MiraMist type nebulizer (Mira Mist CE, Burgener Research Inc., Mississauga, ON, Canada), a model AS 421 autosampler and Analyst 5.4 operating software for data acquisition.

Quantification was performed by interpolation using analytical curves with four standard solutions for calibration. Cd, Cu, Fe, Mn, Ni, Pb, and Zn solutions were generated by dilution in 1000 mg $L^{-1}$ SpecSol stock standard solutions (Quimlab Química & Metrologia®, Jardim California, Jacareí, SP, Brazil) until the desired concentrations were obtained using ultrapure water obtained from an Elga model PURELAB Ultra Analytic (Elga, High Wycombe, Bucks, UK). The instrument operating conditions, detection limits, and method recovery are presented in the supplementary material (Supplementary Table S1).

The SEM calculation was performed by summing the concentrations of trace metals (Zn, Pb, Cd, Ni, and Cu in µmol $g^{-1}$), which may cause toxicity and form more stable monosulfides than Fe monosulfide, which essentially composes the AVS. The SEM/AVS ratios were also shown to always have more AVS than SEM, indicating that the samples had

enough AVS to form monosulfides with all concentrations of Zn, Pb, Cd, Ni, and Cu. Thus, there is no risk of these metals becoming free in solution in the interstitial water of sediments.

### 2.5. Total Organic Carbon and Granulometry

To determine granulometry, 1.5 g of wet sample and 40 mL of dispersing agent (sodium hexametaphosphate 40 g $L^{-1}$) were added to a 50 mL centrifuge tube and stirred for 24 h. After stirring, the particle size was determined using a laser particle analyzer (CILAS Shimadzu, model 1064, Shimadzu, Tokyo, Japan). The obtained results were entered into the Gradistat version 8 program to obtain particle size classification. To analyze the total organic carbon, about 0.02 g of decarbonated dry pellet (1 M HCl, stirred for 16 h) were weighed.

The determination was performed on the Shimadzu TOC-V analyzer using the SSM-5000 A module for solid analysis. To monitor the potassium hydrogen phthalate standard (certified concentration equal to 47.1%), the average concentration of the standard equal to $51.17 \pm 0.02\%$, corresponding to 108.6% recovery.

### 2.6. Adverse Effect Index

The index was calculated using effect range-low (ERL), developed by Long et al. (1995). The effect range low (ERL) and effect range medium (ERM) definitions were based on strongly bound extraction [31]. However, the current study also compared the potentially bioavailable fraction with the safety reference levels since this reactive phase represents the risks associated with biota exposure to bioavailable metals [32].

This index was developed by combining the concentration of metals in sediments with biological effects, representing the concentration above which effects on adjacent biota were observed [33]. The calculation was performed using:

Equation (1):

$$AEI = Me/ERL \tag{1}$$

where:

AEI = adverse effect index
Me = metal concentration
ERL = effect range-low

If $AEI \leq 1$, the metal concentration in the sample is not sufficient to produce adverse effects on nearby biota. If $AEI \geq 1$, the metal concentration in the sample could produce adverse effects [33].

### 2.7. Data Analysis

Correlations between the physical and chemical characteristics of the water and sediment were tested for each area individually using Spearman's nonparametric correlation test. All tests were performed using Statistica software version 8.0.

### 3. Results

In Sepetiba Bay, at the MID, the pH tended to increase gradually as the samples were collected farther from the river (Table 1). Dissolved oxygen (DO), temperature, and conductivity did not change significantly between the samples. At the SFC, the pH was stable among the samples. Dissolved oxygen tended to decrease as samples were collected farther from the river. Conductivity also tended to increase (samples 1 to 4) relative to the furthest samples (samples 8 to 12), while samples 6 and 7 were low compared to others in a similar location.

In the Guanabara Bay (GB) region, the pH, dissolved oxygen, and conductivity tended to increase as the samples were collected farther from the river (Table 2). The temperature slightly increased in relation to the other factors. At the MR, the pH increased noticeably as the samples were collected at points farther from the river, as did dissolved oxygen and conductivity. The temperature decreased proportionally with an increase in the other measured factors.

**Table 1.** Physicochemical parameters of surface water at sampling points in Sepetiba Bay, RJ.

| Madeira Island | | | | | |
|---|---|---|---|---|---|
| Sample | pH | OD (mg L$^{-1}$) | OD (%) | Temperature (°C) | Conductivity (mS cm$^{-1}$) |
| 1 | 7.2 | 4.7 | 67.9 | 23.5 | 47.9 |
| 2 | 7.6 | 4.5 | 59.8 | 23.2 | 48.0 |
| 3 | 7.7 | 4.3 | 60.4 | 23.2 | 48.2 |
| 4 | 7.2 | 4.5 | 63.0 | 23.3 | 48.2 |
| 5 | 6.2 | 4.4 | 62.4 | 23.2 | 50.0 |
| 6 | 7.7 | 4.3 | 61.0 | 23.2 | 48.3 |
| 7 | 7.8 | 4.4 | 62.6 | 22.9 | 48.0 |
| 8 | 7.9 | 4.5 | 63.5 | 23.3 | 48.3 |
| 9 | 7.9 | 4.7 | 67.4 | 23.3 | 48.3 |
| 10 | 7.9 | 4.9 | 68.9 | 23.5 | 48.6 |
| 11 | 7.9 | 4.8 | 66.7 | 23.4 | 48.5 |
| 12 | 7.9 | 4.6 | 65.0 | 23.4 | 48.5 |
| São Francisco Chanel | | | | | |
| Sample | pH | OD (mg L$^{-1}$) | OD (%) | Temperature (°C) | Conductivity (mS cm$^{-1}$) |
| 1 | 6.8 | 10.8 | 126.2 | 22.8 | 3.0 |
| 2 | 6.8 | 8.9 | 110.2 | 23.6 | 12.2 |
| 3 | 6.8 | 8.9 | 106.5 | 23.1 | 28.8 |
| 4 | 6.8 | 7.9 | 103.0 | 23.4 | 23.9 |
| 5 | 6.8 | 6.4 | 106.7 | 23.2 | 47.0 |
| 6 | 6.8 | 7.8 | 97.9 | 23.0 | 18.3 |
| 7 | 6.8 | 7.3 | 92.9 | 23.3 | 24.1 |
| 8 | 6.8 | 7.3 | 102.2 | 23.6 | 48.7 |
| 9 | 6.8 | 7.3 | 107.8 | 23.6 | 49.8 |
| 10 | - | 7.3 | 97.8 | 23.0 | 48.8 |
| 11 | 6.8 | 7.0 | 97.8 | 23.9 | 44.9 |
| 12 | - | 7.2 | 103.4 | 23.3 | 48.4 |

**Table 2.** Physicochemical parameters of surface water at sampling points in Guanabara Bay, RJ.

| Iguaçu River | | | | | |
|---|---|---|---|---|---|
| Sample | pH | OD (mg L$^{-1}$) | OD (%) | Temperature (°C) | Conductivity (mS cm$^{-1}$) |
| 1 | 6.8 | 2.4 | 30.9 | 27.0 | 2.0 |
| 2 | 6.9 | 2.7 | 34.4 | 27.3 | 2.8 |
| 3 | 6.6 | 1.8 | 23.0 | 27.5 | 0.6 |
| 4 | 7.5 | 4.8 | 61.3 | 27.8 | 7.9 |
| 5 | 7.2 | 3.0 | 38.5 | 27.6 | 5.4 |
| 6 | 7.3 | 5.3 | 68.4 | 28.4 | 5.5 |
| 7 | 7.4 | 4.8 | 61.1 | 27.6 | 6.6 |
| 8 | 7.5 | 5.7 | 71.6 | 27.4 | 7.4 |
| 9 | 7.9 | 7.9 | 100.0 | 27.5 | 9.0 |
| 10 | 7.9 | 8.2 | 104.2 | 27.7 | 9.0 |
| 11 | 7.9 | 7.8 | 98.6 | 27.6 | 9.2 |
| 12 | 8.7 | 11.5 | 147.0 | 27.8 | 11.5 |
| Meriti River | | | | | |
| Sample | pH | OD (mg L$^{-1}$) | OD (%) | Temperature (°C) | Conductivity (mS cm$^{-1}$) |
| 1 | 7.2 | 0.00 | 0.0 | 30.3 | 8.1 |
| 2 | 7.2 | 0.00 | 0.0 | 30.8 | 10.7 |
| 3 | 7.4 | 0.00 | 0.0 | 28.8 | 29.1 |
| 4 | 7.5 | 0.40 | 0.1 | 30.8 | 23.7 |
| 5 | 7.7 | 1.6 | 0.2 | 33.1 | 23.0 |
| 6 | 8.3 | 3.8 | 0.6 | 28.7 | 42.1 |
| 7 | 8.5 | 4.5 | 0.7 | 28.2 | 44.0 |
| 8 | 8.6 | 5.0 | 0.7 | 27.7 | 46.5 |
| 9 | 8.4 | 3.8 | 0.5 | 27.6 | 45.3 |
| 10 | 8.4 | 3.6 | 0.5 | 27.2 | 45.8 |
| 11 | 8.7 | 6.0 | 0.9 | 27.7 | 45.4 |
| 12 | 8.7 | 5.6 | 0.8 | 27.6 | 45.8 |

The particle size in the SFC was generally smaller than that of the others, which presented a relatively fine fraction close to 100% (Figure 2), except at sampling point 5. The other monitoring of the Sepetiba Bay (SB) spot (MID) presented relative to fine fractions between 90% and 100%. A similar trend was observed in the BG, with fine fractions ranging from 90 to 100%.

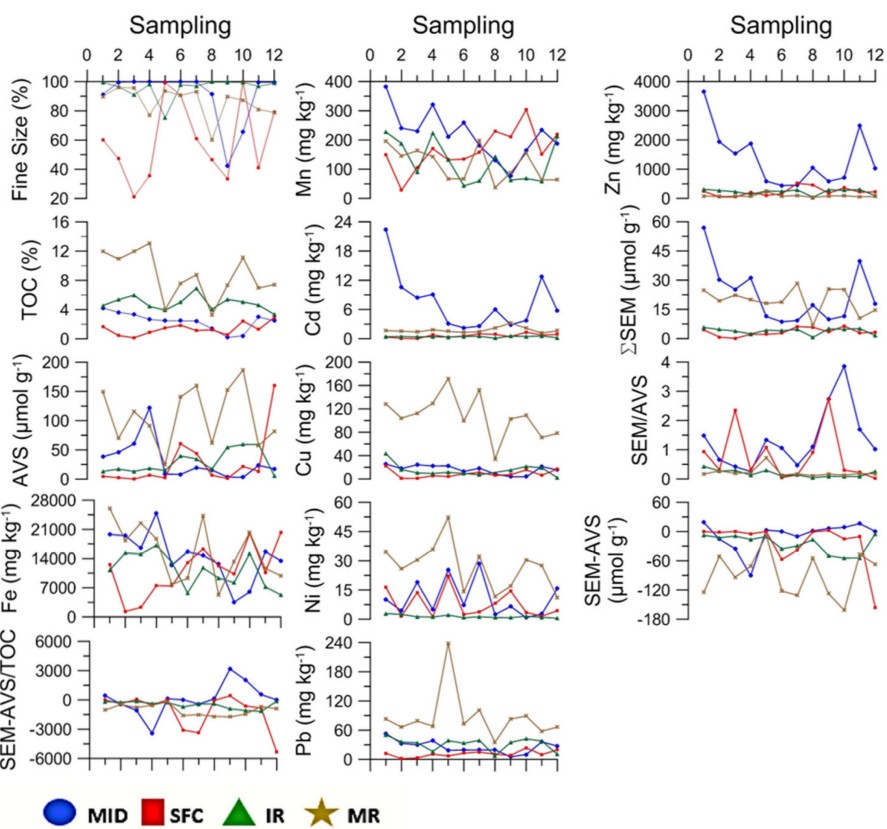

**Figure 2.** Characterization of surface sediment samples collected in Sepetiba and Guanabara Bays, RJ. MID: Madeira Island; SFC: São Francisco Channel; IR: Iguaçu River; MR: Meriti River; SEM: Simultaneously extracted metals; AVS: acid volatil sulfide; TOC: total organic carbon content.

Regarding TOC, SB presented generally lower and more constant values of total organic carbon, whereas GB presented higher variability and generally higher contents comparatively. MID presented a decrescent trend, with a substantial increase in the last two samples; IR and MR presented the highest TOC contents, indicating that Guanabara Bay is a more eutrophicated environment than Sepetiba Bay.

MID presented a higher amount of AVS in most samples, high variability between samples, and high AVS values at 1–4 sampling points, with a decreasing trend in the subsequent samples. The SFC presented most results close to zero, with only samples 6, 7, and 12 having higher values. In Guanabara Bay, MR showed the highest concentrations of AVS, with half of the samples having concentrations above 100 μg mol g$^{-1}$, compared to the IR points, which did not show any concentration above 100 μg mol g$^{-1}$. However, the results suggest that Guanabara Bay is more anoxic than Sepetiba Bay.

Reactive iron concentrations in the MR were high and constant between samples 1 and 4 compared to the other samples, while IR had the highest values. In Sepetiba Bay, MID showed predominantly high reactive iron values, with a drop in samples 9 and 10, and the SFC samples were low in the first four samples, with a slight increase from sample 5 onwards, which remained constant. Iron and manganese presented oscillations in the concentration samples from all locations.

For trace metals, cadmium concentrations were higher in the MID, whereas the other samples presented consistently low concentrations, as were Zn concentrations, which were

higher in the same region, with a peak at sampling point 1. The results for Cd and Zn were expected, due to the contamination by Cd and Zn residues discussed by Monte et al. (2015). These authors found concentrations of Cd and Zn between 0.84 and 17.7 mg/kg and 124 and 3854 mg/kg, respectively.

For copper, the MR samples presented higher concentrations than those in the other areas. Ni concentrations were highest in MR, with a peak at sampling point 5; however, MID and the SFC showed median concentrations with similar behavior. IR showed low concentrations. The lead contents in MID, the SFC, and the IR were consistently low, whereas the MR presented the highest lead concentrations, with sampling point 5 being the highest. The same pattern was observed for Ni.

The IR and SFC presented lower and constant SEM values, while MID presented the highest values in the first samples, comparatively lower values between 5 and 10, and then higher values at sampling point 11. The MR presented constant results for samples from 1 to 10, dropping only at the sampling points of the last transects (8–12 sampling points).

On the ratio between SEM and AVS, the GB region samples were theoretically under 1, with no toxicity. The SB region samples presented values over 1, with the IR samples 1, 5, and 10 and the MR samples 3, 5, and 9 exceeding the toxicity threshold. All the other quantified samples were close to 1. In the Sepetiba Bay, specifically the MID samples, there is a possible correlation between Cd and Zn concentrations with $\sum$SEM, as the samples follow a similar trend.

The SEM-AVS/fTOC ratio was higher at some sampling points in SB with uncertain toxicity, and sampling point 9 of MID showed probable toxicity to the biota. However, at GB, all sampling points in the two studied areas were classified as "no toxicity," suggesting a high degree of eutrophication in this environment [26].

*Correlations between Metals Concentrations and Possible Complexants*

At the MID, TOC was significantly correlated with Cd r = 0.69 and Cu r = 0.89, Pb r = 0.87 and Zn r = 0.68. Reactive Fe significantly correlated with Cu r = 0.84 and Pb r = 0.81. Fe (from SEM) was significant for AVS r = 0.79, Pb r = 0.83, and Zn r = 0.59. Mn (from SEM) was correlated with AVS r = 0.59, Pb r = 0.87, and Zn r = 0.73. Silt content was significantly correlated with Cu r = 0.73, Mn r = 0.61, and Pb r = 0.66 (Spearman, $p < 0.05$, n = 12).

SFC showed similar correlations with MID. In SFC, TOC was significantly correlated with sand r = −0.78; silt r = 0.79; AVS r = 0.72; r = 0.82; Cd r = 0.61; Fe r = 0.84; Cu r = 0.73; Mn r = 0.57; Pb r = 0.85 and Zn r = 0.69. Fe was significant with Cd r = 0.77; Cu r = 0.66; Mn r = 0.66; Pb r = 0.95 and Zn r = 0.80. Mn correlated with Cd r = 0.66 and Pb r = 0.63. AVS was positively correlated with Fe r = 0.63 and Pb r = 0.57, whereas the AVS-SEM/fTOC ratio was negatively correlated with Fe r = −0.60 (Spearman, $p < 0.05$, n = 12).

In Guanabara Bay, the regulators of bioavailability, such as TOC, Fe, Mn, AVS, and grain size, were similar to those found in SB; however, the number of correlations was lower than in the two areas of Sepetiba. In the Iguaçu River, TOC presented correlations with sand r = −0.78; Silt r = 0.79; AVS r = 0.72; Cd r = 0.61; Cu r = 0.73; Fe r = 0.84; Mn r = 0.57 and Pb r = 0.85. Fe was significantly correlated with Cd r = 0.83; Cu r = 0.78; Mn r = 0.78; Pb r = 0.80 and Zn r = 0.61. AVS was positively correlated with Fe r = 0.63 and Pb r = 0.57 (Spearman, $p < 0.05$, n = 12).

In the MR, TOC presented correlations with Fe r = 0.84; Mn r = 0.80 and Zn r = 0.61. Fe was significantly correlated with Mn r = 0.98 and Zn = 0.78. Particle size showed significant correlations, with sand being correlated with Cu r = −0.70 and Zn r = −0.68, while clay was positively correlated with Cu r = 0.79, Pb r= 0.68, and Zn r = 0.70. For sulfides, the AVS was correlated with Fe r = 0.60; Mn r = 0.58 and Zn r = 0.82, and the SEM-AVS/fCOT ratio showed correlation with Ni r = 0.63 (Spearman, $p < 0.05$, n = 12).

## 4. Discussion

$\Sigma$SEM/AVS ratios in MID and the SFC greater than 1 were observed at most sampling points, mainly in the Saco do Engenho region, indicating the potential bioavailability of metals in this area (Figure 2). The results for the two areas of Sepetiba Bay are in a similar

range to those found by Rodrigues et al. (2017); however, MID point 9 showed results almost twice as high as those found on average in this study and [34].

The region is known for large contamination by Cd and Zn, due to a Zn smelting plant in the region, which became environmentally passive after its closure, which resulted in one of the largest known Cd and Zn contaminations in estuarine regions [18,23,34–36]. Rodrigues et al. (2017) found values between 1.1 and 2.3 for the ΣSEM/AVS ratio at the sampling point on Madeira Island. Additionally, toxicity tests performed by the authors confirmed the existence of toxicity related to exposure to metals present in these sediments, as indicated by the ΣSEM/AVS and ΣSEM-AVS/fTOC ratios.

However, it is known that AVS is responsible for complexing metallic compounds in the environment, other complexants, and organic matter [8]. Therefore, the total organic carbon (TOC) value was used to normalize SEM/AVS ratios, and a toxicity classification was proposed by Di Toro et al. (2005). Thus, it was seen that in Sepetiba Bay, even after normalization, several points of MID and two points in the SFC presented a risk to biota, while in GB, there was no associated toxicity.

This efficiency in removing potentially toxic metals from sediments may result in poor availability for biological incorporation into surface waters [37]. At the same time, this process generates the need to clarify how these elements behave within the sedimentary column, as sediments can retain or release previously accumulated metals, due to their greater or lesser ability to maintain metals in the solid phase. This seems to be a valid concern for Guanabara Bay if significant changes occur in the physicochemical conditions of surface waters due to decontamination programs [27,38].

The estuaries of the Iguaçu and Meriti rivers are among the most impacted areas of the bay, which drain most of the metropolitan region of Rio de Janeiro city under the influence of various anthropogenic activities. The eutrophic system and removal of potentially toxic metals from sediments by complexation with organic matter, among other binders, may result in low availability for biological incorporation into surface waters [20].

Nonetheless, in Guanabara Bay, such ratios were less than one at all sampling points in the two areas, presenting the required AVS concentration to trap metal ions [5]. Machado et al. (2010) observed ΣSEM/AVS ratios of less than 1.1 for the first 10 cm of samples collected in the Iguaçu and Meriti rivers. In addition, they indicated that sulfides are the main controllers of metal availability in the Iguaçu River. Abreu et al. (2016) found SEM/AVS ratios below 1 in several sectors of the GB, including the northwest, which is in agreement with a sulfide-rich anoxic environment.

Guanabara Bay is a eutrophicated environment [8,20,26,32], and in this type of environment, one of the main complexants of metals is organic matter [20,26,27]. The organic carbon reduces the solubility and toxicity of metals [39]. Monte et al. (2019), in a study in the Iguaçu River estuary, attributed organic matter as the main complexant in the region, followed by sulfides and oxides and hydroxides of Fe and Mn.

Sulfide production is linked to the grain size composition of the area and the availability of organic matter. A high concentration of organic matter is observed in environments composed of fine sediment fractions, resulting in anoxic conditions for sulfide production [40].

However, Sepetiba Bay showed higher risks to biota than GB, suggesting that the lower TOC percentages found in this study may have influenced the results. Carvalho et al. (2020), found low TOC levels in SB and attributed them to the sandy grain size and the lower organic load compared to other degraded estuarine systems, such as the GB. According to some studies [34,37,41] (Ribeiro et al., 2013; Rodrigues et al., 2017; Carvalho et al., 2020), the percentage of organic carbon has been increasing over the years in Sepetiba Bay, suggesting the influence of increased input from mangroves and river load in the region [37].

On the other hand, straightening projects of the Guandu River to supply fresh water to the Rio de Janeiro Metropolitan area, associated with the growth of harbor activities, and the consequently increased dredging activity, also contributed to a higher contribution of

fine sediments, especially clay in SB, in the last few years [36], which could contribute to the increase in the percentage of organic matter in Sepetiba Bay.

In this study, TOC was positively correlated with Cd, Cu, Pb, and Zn; a result similar to that found by Fonseca et al. (2013) in SB near the MID points. These results corroborate that organic matter is a complex metal complex in the study area [42]. The $Pb^{+2}$ ion in sediments shows a high affinity for substances, forming stable compounds [42].The positive correlations between, Fe and Mn (from SEM) and the metals, mainly in MID and the SFC, except Ni, presented a major geochemical control of bioavailability in SB, a result similar to that found by Monte et al. (2015), after resuspension of sediments in the region near the MID area.

Maddock et al. (2007) performed a study on AVS in the Iguaçu River and Saco do Engenho (the region near the MID) and found that the GB had lower metal bioavailability due to the higher content of organic matter, in addition to the environment being character-ized by redox conditions, which facilitates the formation of sulfides. Sulfide formation is an important mechanism that regulates bioavailability in aquatic environments by forming complexes with metals [43–45].

However, Monte et al. (2021) discussed the bioavailability of metals in sediments of the Iguaçu and Meriti rivers using HCl 1 M extraction and concluded that the northwestern sector of the bay is severely contaminated by metals such as Cu, Pb, and Zn, which are present at high concentrations in the available phase. According to Freitas et al. (2019), in a study of the bioavailability of metals in SFC, the bioavailability of metals was classified as low. However, it can increase after some processes, such as dredging, which are similar to those found by Monte et al. (2015) in Saco do Engenho in Sepetiba Bay. Both authors attributed the increase in bioavailability after resuspension to the oxidation of sulfides and organic matter.

In eutrophic contaminated environments, such as GB, sulfides are formed that, in the-ory, regulate bioavailability; however, the percentage of metals that are in the bioavailable phase is not measured in AVS extraction. The extraction of AVS in this type of environment may show a low apparent risk to the biota, which does not reflect the local reality, as in the case of the Iguaçu and Meriti rivers. In environments with bioavailable metals, some metals (Cd and Pb) can penetrate tissues as essential elements (Ca, Na, and Mg) due to the exchange capacity between ions or, as in the case of Cu, which can be adsorbed from water as hydrated ions [45].

*Sediment Quality Guidelines and Adverse Effect Index*

Concentrations above the ERL and ERM of Cd and Zn in SB (Table 3) were expected in the region, especially in the MID area. Similar concentrations were found by [18,40], suggesting a potential risk to the biota.

In Guanabara Bay, Cd concentrations exceeded the ERL only in the MR; however, Zn concentrations were above the ERL for both rivers, and above the ERM for the MR, suggesting that the degree of degradation of the Meriti River estuary is higher than that of the Iguaçu River, and the main source of Zn in GB is the discharge of untreated sewage.

In the aquatic environment, Zn occurs mainly in the form of $Zn^{+2}$, the ion shows affinity with organic matter, Fe and Mn oxides and hydroxides, and the fine fraction (Fonseca et al., 2013). In this study, Zn was positively correlated with TOC, Fe, Mn, and clay in both bays, as reported in previous studies in Guanabara Bay and Sepetiba Bay, which found similar correlations [18,32,40,42].

High concentrations of Cu in the GB, mainly in the MR, some above the ERL, were reported by Cordeiro et al. (2015) and attributed to the large input of domestic and industrial sewage in this estuary; however, the Cu found in the study by Cordeiro et al. had the highest percentage in unavailable fractions. In environments with a high concentration of organic matter, $Cu^{2+}$ may have a lower mobility because of the high affinity between the ion and organic matter [41,46]. Moreover, the bioavailability of metals can be controlled

in an environment rich in organic matter because of the development of microorganisms, which can influence the speciation of metals [41].

**Table 3.** Comparison between effect range low (ERL) and effect range medium (ERM) and the concentrations of Cd, Cu, Ni, Pb, and Zn found in surface sediments from Madeira Island (MID), the São Francisco Channel (SFC), Iguaçu River (IR), and Meriti River (MR).

| Metal | Content | MID | SFC | IR | MR | ERL | ERM |
|-------|---------|------|------|------|--------|------|------|
| **Cd** | Max | **22.4** | 1.3 | 0.6 | 3.2 | 1.2 | 9.6 |
| | Min | 2.2 | 0 | 0.1 | 1.1 | | |
| | Average | 7.4 | 0.6 | 0.4 | 1.8 | | |
| **Cu** | Max | 25.4 | 22.3 | 43.5 | 171.6 | 34 | 270 |
| | Min | 3.8 | 1.2 | 2.2 | 34.3 | | |
| | Average | 16.4 | 8.9 | 15.1 | 107.8 | | |
| **Ni** | Max | 28.6 | 22 | 2.9 | **52.2** | 20.9 | 51.6 |
| | Min | 0.8 | 1.3 | 0.5 | 11.2 | | |
| | Average | 10.6 | 7.7 | 1.4 | 27 | | |
| **Pb** | Max | 38.2 | 23.7 | 50.4 | **238.3** | 46.7 | 218 |
| | Min | 5.3 | 1.6 | 6.8 | 35.2 | | |
| | Average | 25.8 | 11.2 | 31.7 | 86.7 | | |
| **Zn** | Max | **3646.3** | **518.6** | 309.4 | **1634.5** | 150 | 410 |
| | Min | 440 | 47 | 41.8 | 379.2 | | |
| | Average | **1360.1** | 231 | 229.6 | **1106.0** | | |

In bold the concentrations above ERL and ERM.

Although Ni concentrations were lower in SB than in the MR, some MID and SFC samples were above the ERL, which is related to the increasing industrial activity in the SB region, in addition to the installation of the Casa da Moeda in 1984, near Sepetiba Bay, where the coins that circulate in Brazil are produced that, and for which Ni is the main source material [40]. However, after 2014, the Ni levels in Sepetiba Bay decreased because of the installation of a reuse water system at the Casa da Moeda [40]. Ni concentrations in the MR were above the ERM and according to Baptista-Neto et al. (2013), the sources of Ni in GB are related to the discharge of untreated industrial sewage and urban activities.

Lead showed higher concentrations in the MR, above the ERL and one spot above the ERM. Pb has no biological function and is harmful to living organisms. Its sources may be natural, such as rocks or anthropic, although in the case of this study the sources of Pb in the GB are linked to surrounding industrial activity and fossil fuel burning [47,48]. On the other hand, the source of Pb in SB is the old Zn smelting, which may be associated with higher concentrations at sampling points 1, 2, and 3. MID and the SFC showed the highest concentrations of Mn, which may be related to the formation mechanism of Mn hydroxides and oxides in the region, as found by [18,49].

The AEI is more sensitive in assessing bioavailability compared to using ERL only [33]. The MR showed the worst results regarding the risk to biota; all metals showed values higher than 1 in more than five samples (Table 4). While in MID, the worst results were obtained for Cd and Zn, which was expected for this area. Some sampling points in the Iguaçu River presented a risk to the biota for Zn, which may be related to the large discharge of domestic sewage [20].

The AEI results showed higher sensitivity compared to AVS and ERL in the study of bioavailability in eutrophic or supereutrophic environments, such as Guanabara Bay. This type of environment is complex and can underestimate the bioavailability of trace metals according to the method used for evaluation; therefore, it is necessary to use the methodologies for decision-making used by environmental agencies.

This study observed the different levels of degradation of GB and SB and their impact on the types of geochemical drivers of metal bioavailability. Important retention mechanisms were observed, such as the availability of organic matter and sulfides, which showed different behaviors according to the level of eutrophication in each environment.

**Table 4.** The adverse effect index found in surface sediments from Madeira Island (MID), the São Francisco Channel (SFC), Iguaçu River (IR), and Meriti River.

| Sample | Cd mg/kg | Cu mg/Kg | Ni mg/Kg | Pb mg/Kg | Zn mg/Kg |
|---|---|---|---|---|---|
| MID | **18.6** | 0.7 | 0.5 | **1.1** | **24.3** |
| | **8.8** | 0.5 | 0.2 | 0.7 | **12.9** |
| | **7.0** | 0.7 | 0.9 | 0.6 | **10.2** |
| | **7.6** | 0.7 | 0.2 | 0.8 | **12.6** |
| | **2.6** | 0.7 | 1.2 | 0.4 | **3.9** |
| | **1.9** | 0.4 | 0.3 | 0.4 | **2.9** |
| | **2.1** | 0.5 | 1.4 | 0.4 | **3.0** |
| | **5.0** | 0.2 | 0.1 | 0.4 | **6.9** |
| | **2.4** | 0.1 | 0.3 | 0.1 | **3.9** |
| | **3.1** | 0.1 | 0.0 | 0.2 | **4.7** |
| | **10.6** | 0.6 | 0.1 | 0.8 | **16.5** |
| | **4.8** | 0.5 | 0.8 | 0.6 | **6.8** |
| SFC | 0.3 | 0.7 | 0.8 | 0.3 | **1.6** |
| | 0.1 | 0.0 | 0.1 | 0.0 | 0.3 |
| | 0.0 | 0.0 | 0.7 | 0.1 | 0.3 |
| | 0.7 | 0.2 | 0.1 | 0.2 | **1.4** |
| | 0.2 | 0.1 | **1.1** | 0.2 | 0.7 |
| | 0.5 | 0.3 | 0.1 | 0.3 | **1.1** |
| | 0.8 | 0.3 | 0.2 | 0.3 | **3.5** |
| | 0.7 | 0.2 | 0.4 | 0.2 | **3.1** |
| | 0.4 | 0.2 | 0.7 | 0.2 | **1.2** |
| | **1.1** | 0.5 | 0.2 | 0.5 | **2.4** |
| | 0.6 | 0.2 | 0.1 | 0.2 | **1.4** |
| | 0.7 | 0.5 | 0.2 | 0.4 | **1.5** |
| IR | 0.3 | **1.3** | 0.1 | **1.1** | **2.1** |
| | 0.4 | 0.5 | 0.1 | 0.8 | **1.8** |
| | 0.3 | 0.3 | 0.1 | 0.7 | **1.5** |
| | 0.3 | 0.3 | 0.1 | 0.4 | 0.9 |
| | 0.3 | 0.3 | 0.1 | 0.8 | **1.7** |
| | 0.4 | 0.3 | 0.0 | 0.7 | **1.6** |
| | 0.4 | 0.2 | 0.1 | 0.8 | **1.9** |
| | 0.1 | 0.0 | 0.0 | 0.1 | 0.3 |
| | 0.4 | 0.5 | 0.0 | 0.7 | **2.0** |
| | 0.4 | 0.6 | 0.1 | 0.9 | **1.8** |
| | 0.5 | 0.6 | 0.0 | 0.8 | **2.0** |
| | 0.2 | 0.1 | 0.0 | 0.2 | 0.6 |
| MR | **1.4** | **3.8** | **1.7** | **1.8** | **9.5** |
| | **1.3** | **3.1** | **1.2** | **1.4** | **7.4** |
| | **1.2** | **3.3** | **1.5** | **1.7** | **8.6** |
| | **1.5** | **3.8** | **1.7** | **1.5** | **7.4** |
| | **1.3** | **5.0** | **2.5** | **5.1** | **5.8** |
| | **1.1** | **2.9** | 0.7 | **1.6** | **7.2** |
| | **1.2** | **4.5** | **1.5** | **2.2** | **10.9** |
| | **1.9** | **1.0** | 0.6 | 0.8 | 2.5 |
| | **2.7** | **3.0** | 0.8 | **1.8** | **10.0** |
| | **1.8** | **3.2** | **1.5** | **1.9** | **9.9** |
| | **1.0** | **2.1** | **1.3** | **1.2** | **3.7** |
| | **1.4** | **2.3** | 0.5 | **1.4** | **5.6** |

In bold, the values above 1 represent a risk to the biota.

## 5. Conclusions

Despite having the highest concentrations of metals in the sediment, the northwestern region of Guanabara Bay presented a lower bioavailability of metals compared to Sepetiba Bay when the AVS/SEM model was used. This may be related to the higher

level of eutrophication, which contributes to higher concentrations of organic matter and, consequently, sulfides, due to the low oxygenation of the water/sediment.

However, the AEI proved to be a good tool in relation to the risks to biota. MID and MR showed values above 1 for some metals. One explanation may be related to the fact that the index does not use complexing agents such as organic matter, but only metal concentrations.

Thus, it is necessary that further studies be conducted in estuaries of different trophic levels, but with large concentrations of metals, to draw a more robust evaluation for the application of models and indices of bioavailability of metals suitable and calibrated for each type of estuary.

**Supplementary Materials:** The following supporting information can be downloaded at: https://www.mdpi.com/article/10.3390/coasts3040019/s1, Table S1: ICP OES operating conditions, detection limits and recoveries obtained for the determination of metals in acid extractions carried out in sediments.

**Author Contributions:** A.P.d.C.R. participated in the sediment sampling, carried out the experiments, and wrote, revised, and edited the manuscript; M.M.P. participated in the sediment sampling, contributed to te experiments and the revision of the manuscript; A.C. contributed to the experiments; T.L.d.S.Q. contributed to the experiments and the revision of the manuscript; R.P. contributed to the experiments; T.C.V. participated in the sediment sampling, contributed to the experiments and the revision of the manuscript; R.A.D., proposed the methodology, participated in the discussion, contributed to experiments and the revision of the manuscript; M.M. designed the project and proposed the methodology, contributed to experiments, and the revision of the manuscript; D.A. proposed the methodology and participated in the discussion; C.d.N.M. applied the statistical analyzes, and prepared the figures and tables, participated in the discussion, and reviewed and edited the manuscript; W.M. participated in the sediment sampling, designed the project and proposed the methodology, participated in the discussion, review and editing of the manuscript, acquired funding, and project administration. All authors have read and agreed to the published version of the manuscript.

**Funding:** This research was developed under financial support from CNPq (Proc. No. 481898/2012-3) and CAPES (Financial Code 001).

**Institutional Review Board Statement:** Not applicable.

**Informed Consent Statement:** Not applicable.

**Data Availability Statement:** The dataset generated during and/or analyzed during the current study is available from the corresponding author upon reasonable request.

**Acknowledgments:** The authors thank CAPES for granting a postdoctoral fellowship to Ana Paula Rodrigues (PNPD) and for granting the scientific initiation of a scholarship for Thatianne Vieira; to FAPERJ, for granting the scientific initiation scholarship to Matheus Pereira; and to CNPq for the granting of scientific initiation scholarships for Tássia Quaresma and Rodrigo Pova.

**Conflicts of Interest:** The authors declare no conflict of interest.

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
