# Peer review of "Bioavailability Assessment of Metals from the Coastal Sediments of Tropical Estuaries Based on Acid-Volatile Sulfide and Simultaneously Extracted Metals"

_2673-964X, doi:10.3390/coasts3040019_

Round 1

Reviewer 1 Report

See attached file

Reviewer 2 Report

Revision of the review article titled (Round ONE):

Title: Bioavailability Assessment of Metals from Coastal Sediments of Tropical Estuaries Based on Acid-Volatile Sulfide and Simultaneously Extracted Metals

For the MDPI Journal: Coasts

General observations:      

  1. Manuscript aims at assessing bioavailability of metals in different aquatic environments using ICP-OES, TOC measurements and various calculative models. Nevertheless, the text is very narrative and sometimes lacks scientific focus.

  2. There are 11 respective authors of the article. This is way to many scientists for the production of the work of this size and scope, in my opinion 8 would be more than enough. There are clear MDPI guidelines on paper authorship and not everybody that had contact with manuscript has to appear in the author list. Please reconsider placing some of the names into the acknowledgment section.

Line by line comments:

  1. Line 88: Please state what is the novelty of this study.

  2. Chapter 2.2. Sampling: You claim sampling campaigns were conducted in 2011 and 2012. If this is correct please explain why it took 10 years to measure them? Also, if you made recent (2023) study of those 10 years old samples I would request additional information on how metal and TOC content changes over time. Are You sure there was no lost, changes, organic matter degradation in 10 yrs period? Please explain.

  3. Figure 2 is of very low quality/resolution. Although letters are visible it is overcrowded and hard to grasp. Please reconsider splitting it into two, for example Figure 1a and Figure 1b or similar.

  4. Table 2 brings data of parameters from 2 sampling point with 1 unit of difference in pH among them. Is there any obvious correlation of this pH difference with metal bioavailability performed in the study? Please, state the difference or otherwise (no difference) in the text.

  5. Lines 351-352: Sentence - These results confirmed that organic matter is a complex metal complex - has no meaning. Please reformulate.

  6. Table 4 is very big and spans over 2 pages so it is very hard to follow numbers and obtain some info. Please consider splitting it.

  7. Lines 450-454: Sentence - Assessment of potential bioavailability in eutrophic environments is complex due to the high level of degradation. In such cases, the combined use of several assessment tools is necessary to obtain a robust result according to the local reality before environmental agencies' decision making – does not belong to the Conclusion, at least not as the last sentence of the entire paper. Please consider placing it into Introduction section or reformulate.

  8. References list seems non uniform with various names appearing underlined, text in bold with no need etc. Please check references list or adjust your citation manager.

Please check the language quality especially of the longer sentences.

Reviewer 3 Report

Overall it is an interesting research work aimed at evaluating the potential bioavailability of trace metals using the relationship between metals, AVS, and organic carbon present in the surface sediments of several bays in Brazil, historically known for environmental impacts related to trace metal contamination.

In line 76, the authors could clarify a little more what types of companies are the most significant, in order to better understand what types of activities or toxic discharges are present in the area.

On line 137 and throughout the document, review the chemical nomenclature ("H2S" requires its subindex).

Section 2.3 is not clear: How were the vapors collected? describe the equipment used in more detail.

Why did the quantification focus specifically on Cd, Cu, Fe, Mn, Ni, Pb, and Zn? I would like to know why other metals were not included in the study.

In line 243 of the discussion section, it is important to contrast the numerical values of the studies, e.g., what was the value reported by Monte et al? It is not correct to make a comparison without first indicating which values we are talking about.

Why does line 266 indicate that there was a significant correlation if the p-value was greater than 0.01?

In line 428, how can the authors deduce that Important retention mechanisms were observed, if there is no mention of them, nor is it clearly evidenced in the results obtained? 

The conclusions should be accompanied by some relevant numerical values found in the study.

Conclusions should be rewritten to make them more direct and focused on the objectives of the study.

The authors should perform a grammatical review of the entire document, and verify the placement of commas.

Round 2

Reviewer 1 Report

The authors complied with the recommendations required by me. It can therefore be published in present form.

Reviewer 2 Report

I have no additional comments to the authors.

Reviewer 3 Report

The authors resolved my major concerns about the manuscript. This version is suitable for publication in the journal.